# An Investigation of Soundscape Factors Influencing Perceptions of Square Dancing in Urban Streets: A Case Study in a County Level City in China

**DOI:** 10.3390/ijerph16050840

**Published:** 2019-03-07

**Authors:** Jieling Xiao, Andrew Hilton

**Affiliations:** Birmingham School of Architecture and Design, Birmingham City University, Birmingham, B47BD, UK; andrew.hilton@bcu.ac.uk

**Keywords:** square dancing, soundscape, public spaces, acoustic territory, enjoyment, appropriateness

## Abstract

Square dancing is a popular music-related group physical exercise for health benefits in China mainly participated by mid-aged women and elderly people. This paper investigates the soundscape and enjoyment of the square dancing in urban streets through a case study in Lichuan, a county level city in southwest China, in December 2017. It examines the impact of gender, age, participation and places on perceptions of square dancing soundscape. Two sites along two main urban streets in the city were selected to conduct onsite investigations where residents spontaneously perform square dancing on a daily basis. Ethnographical observations were conducted to identify the social-physical features and sounds of both sites during the dance and without dance. Sound pressure measurements (LAeq and LAmax) were also conducted under the two conditions. An off-site survey was distributed through the local social media groups to understand residents’ everyday experiences and perceptions of square dancing in the city; 106 responses were received for the off-site survey. T-tests and Chi-squared tests were used for statistical analysis of the survey data. The results show gender does appear to be a factor influencing the regularity of participation in square dancing, with a bias towards more female participants. Participation frequency of square dance has an impact on the enjoyment of square dancing. There is no correlation between the dislike of watching square dancing, or dislike of the music and a desire to restrict locations for square dancing.

## 1. Introduction

In the last decade, square dancing (known as grannies’ dance) has gained vast popularity in Chinese cities [1,2]. Square dancing is a physical exercise combined with dancing movements in public spaces. There are three elements that make a square dancing scene: dance leader(s), music, a group of dancers in a paved urban space [3]. The dancing forms vary from gymnastic exercises to folk dance and disco. The music used for square dancing is edited with a plain soundtrack of drumbeats in similar rhythms. Often, the moves are made easy to follow the music. It is believed that the rise of square dancing originated from the historical folk dance, but more recently, Mao’s promotion of sports to enhance people’s health in China in the 1950s. The majority of people who practice square dance in China are middle-aged women and elderly people (both female and male). Square dancing serves as a social activity for people to enjoy themselves with low cost and maintain health. Estimated by the news centre of Chinese Central Television, over 100 million people in the country participate in square dancing everyday. It has become an undefeatable force in China to better understand the impacts and mechanism of square dancing to meet the demand of its large growing elderly population. Physical activities have been proven to bring health benefits, preventing chronic diseases and premature death [4]. The positive impacts of square dancing perceived by its participants are more than physical health benefits [5]. In a survey with square dancing participants in Shanghai, more than 60% of people believed square dancing enlarges their social networks and reduces the feeling of loneliness [6]. Group dancing serves as a tool to form a sense of community and generate collective experiences [7]. 

Criticism has been raised regarding square dancing for causing noise pollution [5,8]. The operating sound level during the dancing is often beyond the limits set by the national environmental noise control regulation (GB22337-2008) for urban streets - 70 dBA during daytime (10:00 pm–6:00 am) and 55 dBA at nighttime (6:00 am–10:00 pm) [9]. Responding such issues, the General Administration of Sport of China published legislations to force local authorities to manage the dancing groups, limit the time for dancing, create more activity spaces and control the noise and content of music used [10]. This has drawn attention to the sonic spatiality of public spaces that support music-related activities whilst not causing annoyance. Sounds generated by music-related activities such as street performance, taichi and skateboarding, are a common and important part of the sonic environment in the urban context [11]. A better understanding of the interrelationship between the music, perceptions of square dancing and place is needed to further promote this physical activity in open public spaces.

The soundscape concept provides a conceptual framework to understand the square dancing phenomenon. International Organization for Standardization defines soundscape as the acoustic environment perceived/understood or experienced by a person or people in context [12]. The sensation, interpretations and responses to the perceived acoustic environment play a central role in the concept. In recent studies of soundscapes in outdoor public spaces, activities have been considered as a key role in creating urban soundscapes whilst contextualizing the way people evaluated the perceived sound environment [8,13,14]. This dual effect has drawn attention to look at the activity-soundscape relations in spaces [15]. Positive soundscapes are considered to provide an acoustic environment to mediate site-specific activities and increase pleasantness of using the space [16]. The perception of sound is also considered as an aesthetic sensation that people constantly examine the pleasure in the listening process [17]. Expectations, familiarity and level of social interactions are considered key factors influencing soundscape evaluations of performed activities [18]. 

The spatial distribution and types of spaces are in the core debate of regulating square dancing. However, few studies have examined square dancing in urban streets. In many cases, for convenience and space shortage, people choose to dancing on busy street median strips and sidewalks [3]. A street has different layers forming a spatial sequence from buildings on the side to sidewalks for pedestrians and lanes for cyclists and vehicles. Sounds perceived from waling in urban streets are important to the kinaesthetic experiences of everyday cities [19]. Unlike parks or squares which are designed purposefully for active physical activities, sidewalks on urban streets seem to be less functional and appropriate for square dancing. Square dancing in inappropriate places will be perceived as noise and cause stress to its perceivers and decrease the quality of life. Regular exposure to uncontrollable noise in urban environments will cause after-exposure long-term health issues associated with depression and sleeping problems [20]. Thus, the impacts square dancing have on the acoustic environment in urban streets and whether types of places have an impact on people’s perceptions of square dancing in everyday experience will be explored in this paper. 

Gender and age play important roles in evaluating loudness and acoustic comfort in public spaces. Significant difference has been found in preferences of soundscape elements in urban spaces among different age groups [21]. Females are more sensitive to sounds than males [22]. The majority of square dancing participants are mid-aged females [10] which indicates favouritism of this activity in women and a certain age range. However, from the audience perspective in a recent study, female audiences rated lower on their evaluations of acoustic comfort in parks when square dancing took place [23]. Previous studies suggest the length of time people spent in a place and visiting frequencies influence their evaluations of acoustic comfort [24,25]. In the case of urban streets where the main function is conceived as transit and movement, the time people spent at a particular point is very limited. Thus, durations of stay are not essential in the context of square dancing in urban streets. Instead, participation of square dancing might have an impact on perceptions of square dancing and associated acoustic comfort in urban streets. 

Square dancing brings dual perceptual evaluations of the space from both perspective of the audience and participants. Compared with non-music-associated activities, music-associated activities in urban spaces attract audiences through aural-visual interactions [16]. In particular, larger group activities (n > 6) have more significant influences on pedestrians’ behaviours and the time they stop to watch [11]. The associations of sounds in streets are argued as the key to initiate the social interactions between the space and the moving body [19]. Music might have played an important role in the spontaneity of square dancing, breaking the boundaries between audience and dancers. Sound pressure levels have been used to as a key objective factor to examine acoustic comfort in public spaces. People in public spaces rated low acoustic comfort when the sound pressure level exceeds 73 dBA [24]. A recent study on square dancing in a park in Harbin China measured the sound pressure level difference in the park when square dancing occurred as well as surveyed the objective loudness perceived by visitors [23]. The results reveal the music of square dancing has no significant impact on users’ subjective evaluations of soundscape comfort in the park. The sound pressure level difference in the case was only 3 dBA and the average sound pressure level was less than 70 dBA. However, in urban streets where the background noise level is relatively higher than in parks and squares, it is questionable whether the sound pressure level and perceptions of the square dancing music are similar.

Thus, this paper aims to answer the questions: (1) What influence does square dancing have on the acoustic environment in urban streets? (2) How do gender, age and participation influence the enjoyment of square dancing? (3) Are urban streets perceived appropriate for square dancing to occur? Mixed methods of onsite acoustic measurements, observations and off-site surveys were used to conduct the study in two sites in a selected case city.

## 2. Methods

### 2.1. Selection of Case Study

In metropolitan cities, like Beijing and Shanghai, square dancing has been banned in main streets and certain public spaces to deal with the noise compliance and public order. In order to investigate influences of square dancing in urban streets, a case study was conducted in a county-level city in southwest China (Lichuan) where square dancing is popular and relatively less controlled by the government compared to the metropolitan areas. The city has three main streets running through the urban area in parallel from west to east and a number of squares in front of key infrastructures such as government, hospitals and schools to facilitate the everyday public social activities. Square dancing spreads out in the city particularly along sidewalks in the three main streets. Two sites were selected in the case study city (Figure 1): site 1 along Qingjiang Road (high street) and site 2 along Binjiang Road (riverside). Qingjiang Road is mainly for commercial activities with buildings on both sides as shops, restaurants, hotels and a mixed-use residential complex. However, alongside Binjiang Road, there are mainly residential buildings and offices on one side and the Qingjiang river on the other side. 

Site 1 is located outside a mixed-use residential building with shops on the ground floor. There is a paved space at the front of the building around 120 square metres (18 m × 6 m). A dance team starts dancing around 6:00 pm everyday there with a speaker positioned against the building façade projecting music towards the street. Site 2 is located along the paved riverside walk opposite the city’s media and culture bureau. The site is stepped back from the pedestrian with a paved area around 300 square metres (30 m × 10 m). The speaker was positioned against the retaining wall projecting towards the open space facing the river. 

### 2.2. Onsite Measurements

Data were collected in December 2017, using Castle-Group GA216L Sound Level Metre. During the measurements, the SLM was provided with a wind-screen and located on a tripod at a height of 1.50 m from the ground to reduce the effect of acoustic reflections (refer to [11]). Measurements were taken between 6:00 and 8:00 pm (the peak time for square dancing), conducted 30 min after the start when the size of group is more stable. A-weighted equivalent sound levels (L_Aeq_) will be measured at slow-mode for 1-minute intervals three times in three different positions (as shown in Figure 2 where the red dot represents the location of the speaker and light grey dots represent participants in the square dancing):1 m from the loudspeaker (frontal position)In the middle point of the area occupied by the square dancers (frontal position)At the farthest point of the area occupied by the square dancers (side position).

Rather than noise monitoring, repeated recordings in short intervals are useful to capture the characteristics of the acoustic environment in an urban context [26]. In order to compare with no dancing conditions, a set of three 1-minute recordings were performed in position a and c as taken in in-session measurements to obtain an A-weighted equivalent sound level. The measurements outside the session were conducted between 1:00 and 2:00 p.m. to get a representative acoustic environment in streets for normal everyday city life. Considering the fluctuating sound features of square dancing music, the peak levels (L_Amax_) will also be recorded to reflect on attractions of attentions during the dance. 

### 2.3. Ethnographical Observations

Observations were conducted by the principle investigator onsite along with the measurements in order to map out the settings of square dancing on two sites (Table 1). The purpose was to note the size of the dancing group, built forms of the space and the engagement with the audience.

### 2.4. Off-Site Survey

Unlike previous studies, an off-site survey was conducted to understand how people perceive square dancing in a general sense in their everyday life rather than constrained to the site and temporary conditions. The survey was distributed online through local social media networks between 23 December 2017 and 23 January 2018 in the case study city. In total, 106 responses were received within the month when the measurements were conducted with an age range from twenty years old to seventy. Of those participants 69% were female, 78% were local residents with a further 17% originating from other county level cities and 4.7% from larger provincial cities. The age ranges from 20 through to 70, with 23% aged 20–30, 14% aged 30–40, 30% aged 40–50, 30% aged 50–60 and 3% aged 60+. Five questions were designed to gather data for statistic analysis on interrelationships between gender, age, participation, place and enjoyment of square dancing:
(1)Do you enjoy watching square dancing?(2)How often do you participate in square dancing?(3)How would you describe square dancing?(4)Do you find the square dancing music unpleasant?(5)What places are appropriate for square dancing?


Participants were given options to choose from for each question. Those options were developed based on the existing perceptual indicators, which focused more on the comfort and unique soundscape features of square dancing. Thus, three bipolar psycho-acoustic descriptors were chosen from previous studies [27,28,29] as multiple options to select in the survey for subjective perceptions of square dance: annoying-pleasing, noisy-rhythmic, boring-interesting. 

### 2.5. Ethics

The study was conducted in accordance with the Declaration of Helsinki, and the protocol was approved by the Research Ethics Committee in Faculty of Arts, Media and Design at Birmingham City University on 20th December 2017. The survey is anonymised and has gained consent from the participants to use the data for academic research and publications.

## 3. Results

### 3.1. The Objective Loudness and Acoustic Environments

Mean values of sound pressure level measurements at the identified points were used to assess the change of objective loudness in urban streets during square dance compared to midday normal everyday conditions (see Table 2). The background noise level between 1 and 2:00 pm on site 2 is 2–5 dBA lower than at site 1. Significantly increased sound pressure levels were found at point a during the dance on both sites compared to midday without square dance. The most dominant sounds observed on both sites during the square dance were the square dance music from the audience side whilst the sound of traffic was dominant during midday. There is a significant drop from the speaker (point a) to the edge (point c) of the dancing group on both sites: 10 dBA difference at site1 whilst 30 dBA difference at site 2 during the dance. However, no significant difference was found at point c compared to normal conditions with only 2 dBA increase in site 1 and 5 dBA in site 2. 

Although the background noise level at site 1 was higher than that at site 2, the sound pressure level of the broadcasting music during the dance at site 2 was much higher (17 dBA more) than at site 1. The scale of space and intensity of commercial activities onsite seem to have an impact on the background noise and loudness of the music broadcasted to reach to the edge of the group. The space of site 2 was twice the size of site 1 and without any solid construction within 30 m from the edges. Peak levels during the dance on both sites were found to be only 5–6 dBA different from the continuous equivalent levels. It seems unlikely that people will be attracted by the incident sound from the speaker. 

### 3.2. Perceptions of Square Dance and Square Dance Music

Based on the survey data collected, a standard 95% confidence interval calculation was conducted to determine the true proportions of participants’ responses to different factors. The true proportion of participants in the survey who regularly (more than once a week) participate in square dancing is between 24% and 41.9%. Over half of participants in the survey never practiced square dance. There is a variation in preference of watching square dance and listening to the square dance music among different age groups (see Figure 3). 

A series of T-tests were carried out to determine factors influencing the enjoyment of square dancing. A correlation was found between those who described square dancing in negative terms and those who find the music unpleasant (see Table 3). This correlation also held true between those not enjoying watching square dancing and those finding the music unpleasant 

It can be concluded that with at least a 95% degree of certainty that perceived comfort of the music has an influence of whether people enjoy watching square dancing (see Table 4). Furthermore, there was a significant difference between the mean ages of those who enjoy watching square dancing and those who do not enjoy watching square dancing (see Table 5). This correlation with mean age also held true for the unpleasantness of the music (see Table 6) and the regularity of participation (see Table 7).

However we found *no significant difference* in gender associated with these preferences. Using a Chi-squared test (χ^2^) we can conclude that with a 95% degree of certainty, gender has no significant impact on the enjoyment of watching square dance (χ^2^ = 3.80 which is not > 5.991 for 95% with 2 degrees of freedom Table 8), of finding the music pleasant (χ^2^ = 6.20 which is not > 7.815 for 95% with 3 degrees of freedom, Table 9).

However, our research did find some correlation between gender and the regularity of participation in square dance activities (χ^2^= 4.77 which is > 3.841 for 95% with 1 degree of freedom, Table 10), suggesting that females are more likely to participate more regularly.

From the t-test data it can also be concluded to at least the same degree of certainty that age does in fact have a significant impact on the enjoyment of and participation in square dance. However, the distribution of online surveys might have an impact on the age groups who have access to the survey. This needs further investigation through a larger sampling survey through other methods.

A strong correlation was found between those who participate regularly with both the perceived unpleasantness of the music (see Table 11) and the enjoyment of watching square dance (see Table 12)

However, there are 30% who occasionally are annoyed by the square dance music. The reasons for the occasional annoyance might come from personal conditions which are difficult to predict or control. Unlike previous literature, the overall perceptions of square dance in the survey are not negative particularly from the audience’s perspectives (32.4% of those who rarely or never participate in square dance identified as not enjoying watching square dance). The music of square dance plays important roles in their perceptions where rhythm was the most frequently selected psycho-acoustic descriptor (see Figure 4). The percentage (based on 95% confidence intervals) of people who find square dancing music pleasant is between 63.6% and 78.4%.

### 3.3. Perceptions of Appropriate Places for Square Dance

The majority of respondents (82%) suggested that external spaces are most appropriate for square dance, with only 18% suggesting that the interior spaces of gymnasium are most appropriate (see Figure 5). The percentage of participants (based on 95% confidence intervals) in the survey who think interior spaces are most appropriate for square dancing is between 11.8% and 24%. The percentage of participants who identified themselves enjoy the square dancing music and prefer square dance to occur in public urban spaces (i.e., urban squares and streets), rather than parks or community open spaces which are usually in gated enclaves, is between 27.6% and 46.2%. However, only a few considered sidewalks in urban streets like the case studies are appropriate for square dance. Park and squares are most frequently selected places which are appropriate for square dance to occur.

A range of T-tests and Chi Squared Tests were conducted to ascertain any correlation between spatial preferences for the performance of square dance. The majority of respondents (82%) suggested that external spaces are most appropriate for square dance, with only 18% suggesting that the interior spaces of gymnasium are most appropriate. Using a Chi Squared Test, there was no correlation between those who do not identify as enjoying square dance and a preference for interior spaces (see Table 13, χ^2^= 3.28 which is not > 5.991 for 95% with 2 degrees of freedom).

Using a T-test we found no correlation between those who described the music as being unpleasant and the preference for square dancing to be outside public urban areas (t = 0.8518 which is not >1.96 for the 95%, Table 14). 

However there was a tentative correlation (90–95% significance) between those who describe square dancing in negative terms (boring, noisy or annoying) and a preference for dancing to occur outside public urban spaces (t = 01.9082 which is not >1.96 for the 95% but is >1.645 for the 90% see Table 15). 

Using a Chi Squared Test, there was no significant difference between respondents description of square dance and the preference for public urban spaces (see Table 16), suggesting that people do not mind square dancing in urban spaces even if they do not enjoy participating, watching or listening to the music. No significant difference was found between those who participate regularly or rarely and a preference for square dances to occur in public urban spaces (see Table 17).

Thus, participation of square dancing has no influence on perceptions of the physical settings of the soundscape and visual interactions of square dances. There is also no significant difference in preference of spaces for square dance to occur between those who described square dancing in a positive way and those who described it in more negative terms.

## 4. Discussion

### 4.1. Acoustic Boundaries between Square Dancers and Audience in Streets

The square dancing music also redefines acoustic territories in urban streets. Perception of sounds in public spaces is a dynamic process of disintegration and reconfiguration shifting between personal and shared experiences [17]. Sounds in streets create temporary territories for specific activities and attract attention from the outside. However, what defines the boundaries of acoustic territories is arguable. 

The objective loudness is not decisive in defining the acoustic boundaries of square dancing in urban streets. In the case explored, there is only a 2-5 dBA difference at the audience side (edge of the dance group) during the dance which causes no significant contrast of loudness in the streets. The background sound level influences soundscape evaluations in urban open public spaces and a lower background level makes people feel quieter [24].

Although a 17 dBA difference was found near the speaker during the dance, similar sound levels were found at the edge of the dance group around 73 dBA. With a similar background noise level, 73 dBA might be the threshold sound level for participants to engage in the dance with the music. This territory is different from the audience’s. It is suggested that people within 10 meters from edge of the music related activity group often will be attracted to stay and watch [11]. However, this might not be true for square dancing in urban streets. The significant correlation between the enjoyment of square dance music and watching square dance indicates the acoustic boundaries of the audience are set by the audible level of the square dance music. This might also relate to the scale of the space and visual distance between dancers and the dance leaders at the front.

### 4.2. Impacts of Age and Gender on Perceptions of Square Dancing

A previous study found differences in different age groups when evaluating acoustic comforts [21]. Personal preferences of different sounds play an important role in this evaluation. The older age groups in the survey enjoy watching square dancing or listening to the music which reveals a generational gap in preferences of musical sounds and activities in public urban spaces. Most participants in the age range between 20 and 30 rarely practice square dancing. The involvement and appreciation of square dancing seem to have an impact on their perceptions. The social context of the city (being a county level city, there are fewer options of leisure activities whilst closer social relationships) explored might also have impacts on people’s perceptions and participation of square dancing. 

Although square dance in China is commonly perceived as an exercise or leisure for mid-aged women [10], there is no difference in gender of perceiving square dances and the music in the case study. However, the types of square dance music in the city explored are quite singular and mostly based on popular folk music commonly heard on television without gender bias. This might be different if the types of music and dance are more gender targeted.

### 4.3. Impacts of Place Context on Perceptions of Square Dance

The preconceptions of soundscapes in different types of places might have impacts on perceptions of square dance and the music. The perceptions of a park being suitable for physical exercises with low background noise and squares for music-related activities with a loud background sound level have a significant impact on people’s choices of square and parks as the most appropriate places for square dance. However, sidewalks in urban streets as explored in the case are not in preference for square dance. The expectations of space for square dancing to be a large flat area [3] might have influenced their evaluations of street sidewalks for square dancing. However, this is also related to the type and size of square dancing groups occurring frequently in participants’ everyday experiences. 

However, people do not seem to be influenced by the appearance of the physical space when evaluating their perceptions of square dance. No correlations were found between enjoyment of square dance and the choices of places. In the city explored, the time when square dance happens (between 6:00 pm and 8:00 pm) is often after sunset. The visual appearance of the physical space does not have much influence. This might be different in other places where there is a late sunset in some seasons. 

### 4.4. Limitation of this Study

This study explored square dancing soundscape in a singular case which cannot represent the diversity of square dancing in other cities in a different context. The study was conducted in winter where the participation, clothing factor and background noise levels might be different from other seasons. The off-site survey based on people’s everyday experiences retrieved from memories might be different from in-situational experiences. Onsite survey will be needed to explore the in-situ experiences from both perceivers’ and dancers’ views to further investigate the acoustic territories. 

A previous study suggests that one potentially effective solution to control noise from square dancing is to use Bluetooth earphones [10]. The level of social interactions will be reduced in this sense. It might also cause safety issues since no sound signals will be received by other users in the street. Further investigations might be useful to examine this assumption through an initiated Bluetooth earphone aided square dancing.

## 5. Conclusions

It can be concluded from this study the main factor influencing the enjoyment of square dancing was age rather than gender. However, gender does appear to be a factor influencing the regularity of participation in square dancing, with a bias towards more female participants. Participation in square dancing has an impact on the enjoyment of square dancing. Those who participate in square dancing regularly and those who enjoy watching square dancing are more likely to find the music pleasant.

Similarly, those who found square dancing music unpleasant were less likely to participate in or watch square dancing; however, this dislike did not influence their preferences for square dancing to occur outside public urban spaces. 

There is no correlation between the dislike of watching square dancing, or dislike of the music and a desire to restrict locations for square dancing. Likewise, no correlation was found between enjoyment of any aspect of square dancing and the desire for specifically urban spaces.

It can be seen from the results that the enjoyment of the music of square dancing has no significant influence on the preferred spatial location for square dancing. 

## Figures and Tables

**Figure 1 ijerph-16-00840-f001:**
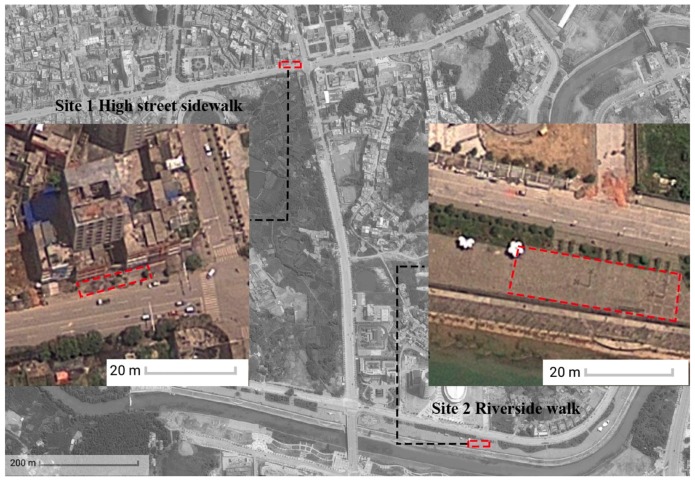
Map showing the two sites in the case study city and the urban morphology of the city.

**Figure 2 ijerph-16-00840-f002:**
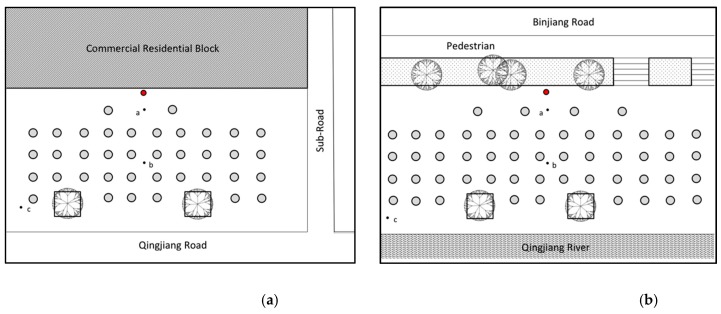
Illustrations of points for measurements onsite during the dance (where the red dot represents the location of the speaker and light grey dots represent participants in the square dancing): (**a**) layout of measuring settings in relation to the speaker at site 1; (**b**) layout of measuring settings in relation to the speaker at site 2.

**Figure 3 ijerph-16-00840-f003:**
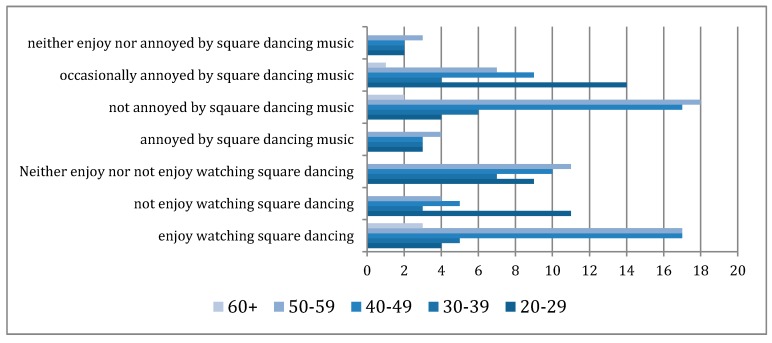
Responses from different age groups on the enjoyment of watching square dancing and the annoyance of square dancing music.

**Figure 4 ijerph-16-00840-f004:**
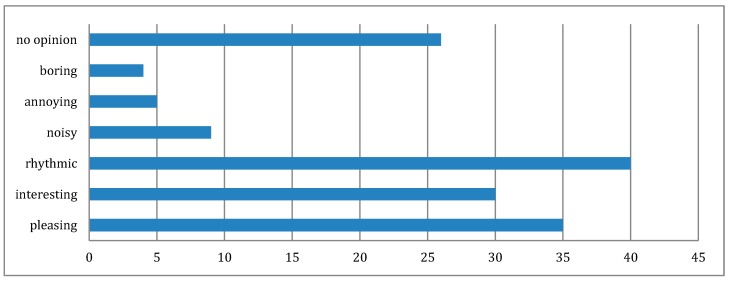
Frequencies of psycho-acoustic descriptors selected by participants in the survey to describe their perceptions of square dancing soundscape.

**Figure 5 ijerph-16-00840-f005:**
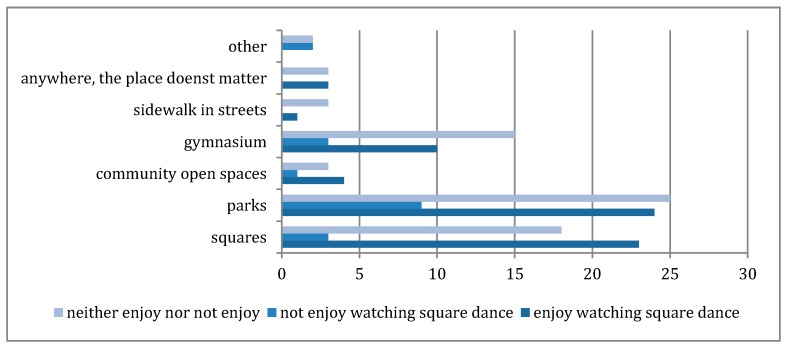
Participants’ responses for appropriate places for practicing square dancing and their preferences of watching square dancing.

**Table 1 ijerph-16-00840-t001:** Observation notes and photos of sites during square dancing at 6–8:00 pm and without square dancing at 1–2:00 pm.

**Site 1 Qingjiang Road (High Street)**			
**Time**	**Observations**		**Photo of the site**
1–2:00 pm without square dancing	Dominant sounds onsite:	traffic	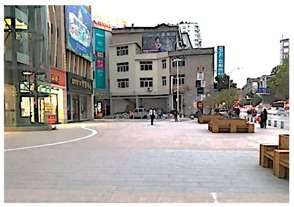
	Other sounds:	people talking, advertisement music from shops on the side	
	Activities onsite:	People walking pass and walking in/out of the shops	
6–8:00 pm during square dancing	Dominant sounds onsite:	square dancing music	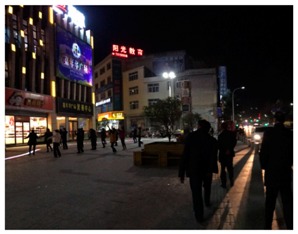
	Other sounds:	people talking, traffic	
	Activities onsite:	square dancing, people walking pass and walking in/out of shops, people watching square dance	
	Size of the dancing group:	started with 12 and increased to 25 after 30 min	
**Site 2 Binjiang Road (Riverside walk)**			
**Time**	**Observations**		**Photo of the site**
1–2:00 pm without square dancing	Dominant sounds onsite:	traffic, people talking	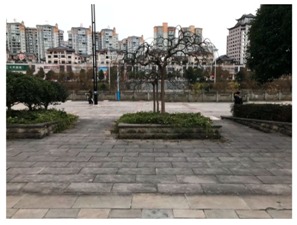
	Other sounds:	music from the other side of the river, people talking	
	Activities onsite:	People walking pass, children play	
6–8:00 pm during square dancing	Dominant sounds onsite:	square dance music	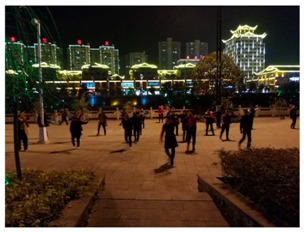
	Other sounds:	Children playing	
	Activities onsite:	people walking pass, people watching square dancing	
	Size of the dancing group:	started with 18 and increased to 45 after 30 min	

**Table 2 ijerph-16-00840-t002:** Measurements of sound pressure levels (LAeq and LAmax) at two sites during and without square dance.

Site	Condition	Point	LAeq (mean)	LAmax (mean)
Site 1 Qingjiang Road sidewalk (High street)	Without square dance 1:00–2:00 pm	a	64.03	74.50
		b	n/a	n/a
		c	71.43	85.63
	During square dance 6:00–8:00 pm	a	87.50	94.20
		b	76.07	83.43
		c	75.10	80.43
Site 2 Bingjiang Road side walk (Riverside)	Without square dance 1:00–2:00 pm	a	59.57	65.40
		b	n/a	n/a
		c	69.03	79.40
	During square dance 6:00–8:00 pm	a	104.43	109.27
		b	93.03	99.73
		c	74.30	79.50

**Table 3 ijerph-16-00840-t003:** Two-Sample t test with unequal variances comparing the description of square dancing with pleasantness of music.

	Description of Square Dancing					
Heading	*n*	Mean	Std. Deviation	Std. Error Mean	95% Confidence Interval	
					Lower	Upper
Describe Square Dancing as Pleasant	78	5.71	1.19	0.053	5.446	5.974
Describe Square Dancing as unpleasant	65	4.83	1.85	0.018	4.38	5.28

Pr (|T| > |t|) = 0.001, t = 3.2885.

**Table 4 ijerph-16-00840-t004:** Two-Sample T test with unequal variances comparing enjoyment of watching square dance with pleasantness of music.

	Enjoyment of Watching Square Dance					
	*n*	Mean	Std. Deviation	Std. Error Mean	95% Confidence Interval	
					Lower	Upper
Describe Square Dancing Music as Pleasant	57	2.42	0.68	0.09	2.2436	2.5964
Describe Square Dancing Music as unpleasant	49	1.98	0.83	0.12	1.7448	2.2152

Pr(|T| > |t|) = 0.004, t = 2.9673.

**Table 5 ijerph-16-00840-t005:** Two-Sample t test with unequal variances comparing Mean Age with enjoyment of watching square dance.

	Mean Age					
	*n*	Mean	Std. Deviation	Std. Error Mean	95% Confidence Interval	
					Lower	Upper
Enjoy Watching Square Dancing Yes	46	47.17	10.32	1.522	44.19	50.15
Enjoy Watching Square Dancing NO	23	35.87	12.03	2.51	30.95	40.79

Pr(|T| > |t|) = 0.0004, t = 3.8547.

**Table 6 ijerph-16-00840-t006:** Two-Sample t test with unequal variances comparing Mean Age with unpleasantness of square dance music.

	Mean Age					
	*n*	Mean	Std. Deviation	Std. Error Mean	95% Confidence Interval	
					Lower	Upper
Unpleasantness of Square Dancing Music Yes	71	44.01	11.36	1.348	41.3676	46.652
Unpleasantness of Square Dancing Music No	15	35.67	12.23	3.158	29.481	41.859

Pr(|T| > |t|) = 0.025, t = 2.4315.

**Table 7 ijerph-16-00840-t007:** Two-Sample t test with unequal variances comparing Mean Age with regularity of square dancing participation.

	Mean Age					
	*n*	Mean	Std. Deviation	Std. Error Mean	95% Confidence Interval	
					Lower	Upper
Regularly Participate in Square Dancing Yes	29	50.86	8.25	1.532	47.86	53.86
Regularly Participate in Square Dancing No	77	39.55	11.65	1.328	36.95	42.15

Pr(|T| > |t|) = 0.0000 t = 5.5849.

**Table 8 ijerph-16-00840-t008:** Chi Squared Test comparing gender to enjoyment of watching square dance.

Enjoyment of Watching Square Dancing	Gender (*n* = 106)		χ^2^	φ
	Male	Female		
Yes	10	36	3.80	0.036
Do not Know	13	24		
No	10	13		

*p* = 0.1495, df = 2.

**Table 9 ijerph-16-00840-t009:** Chi Squared Test comparing gender to unpleasantness of square dance music.

Unpleasantness of Square Dancing Music	Gender (*n* = 106)		χ^2^	φ
	Male	Female		
Yes	8	6	6.20	0.058
Do not Know	4	6		
Occasionally	10	25		
No	11	36		

*p* = 0.1023, df = 3.

**Table 10 ijerph-16-00840-t010:** Chi Squared Test comparing gender to regularity of participation in Square Dance Activities.

Regularity of Participation in Square Dancing	Gender (*N* = 106)		χ^2^	φ
	Male	Female		
Regularly *	6	29	4.77	0.045
Rarely **	27	44		

*p* = 0.029, df = 1, * Participation at least once per month. ** participation less than twice per year.

**Table 11 ijerph-16-00840-t011:** Chi Squared Test comparing regularity of participation in Square Dance Activities to the perceived unpleasantness of the square dance music.

Unpleasantness of Square Dance Music	Regularity of Participation (*n* = 106)		χ^2^	φ
	Regularly *	Rarely **		
Yes	3	11	10.26	0.097
Do not Know	1	9		
Occasionally	8	27		
No	23	24		

*p* = 0.0165, df = 3, * Participation least once per month. ** participation less than twice per year.

**Table 12 ijerph-16-00840-t012:** Chi Squared Test comparing regularity of participation in Square Dance Activities to the enjoyment of watching square dance.

Enjoyment of Watching Square Dance	Regularity of Participation (*n* = 106)		χ^2^	φ
	Regularly *	Rarely **		
Yes	30	16	39.27	0.37
Do not Know	5	32		
No	0	23		

*p* = 0.000, df = 2, * Participation at least once per month, ** participation less than twice per year.

**Table 13 ijerph-16-00840-t013:** Chi Squared Test comparing the description of square dance to the preferred location of Square Dance.

Preferred Location of Square Dance	Description of Square Dance (*n* = 144)		χ^2^	φ
	Positive	Negative		
Public Space	38	9	3.28	0.023
Interior	17	10		
Non Urban	48	22		

*p* = 0.194, df = 2.

**Table 14 ijerph-16-00840-t014:** Two-Sample T-test with unequal variances comparing unpleasantness of Square Dance Music with preferences for urban or non-urban spaces.

	Unpleasantness of Square Dancing Music					
	*n*	Mean	Std. Deviation	Std. Error Mean	95% Confidence Interval	
					Lower	Upper
Preference for Urban Space	47	3.00	1.08	0.157	2.691	3.309
Preference for Non Urban Space	96	2.83	1.13	0.115	2.604	3.056

Pr(|T| > |t|) = 0.3964, t = 0.8518.

**Table 15 ijerph-16-00840-t015:** Two-Sample T-test with unequal variances comparing a negative description of Square Dance Music with preferences for urban or non urban spaces.

	Description of Square Dancing					
	*n*	Mean	Std. Deviation	Std. Error Mean	95% Confidence Interval	
					Lower	Upper
Preference for Urban Space	76	5.70	1.30	0.149	5.666	5.733
Preference for Non Urban Space	148	5.32	1.54	0.1266	5.300	5.34

Pr(|T| > |t|) = 0.05799, t = 1.9082.

**Table 16 ijerph-16-00840-t016:** Chi Squared Test comparing description of square dance to the preferred location of Square Dance.

Description of Square Dancing	Preferred Space for Square Dancing (*n* = 232)		χ^2^	φ
	Urban	Other		
Pleasing	28	37	4.75	0.02
Rhythmic	26	43		
Interesting	19	32		
No opinion	6	19		
Boring	1	5		
Noisy	4	8		
Annoying	0	4		

*p* = 0.576, df = 6.

**Table 17 ijerph-16-00840-t017:** Chi Squared Test comparing preferred location of Square Dance with regularity of participation.

Preferred Space for Square Dancing	Regularity of Participation in Square Dancing (*n* = 143)		χ^2^	φ
	Rare	Regular		
Parks & Other	43	18	2.17	0.0152
Gymnasium	19	8		
Community open Space	6	2		
Urban Squares	28	16		
Urban Streets	3	0		

*p* = 0.704, df = 4.

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
