# Peer review of "An Investigation of Soundscape Factors Influencing Perceptions of Square Dancing in Urban Streets: A Case Study in a County Level City in China"

_ijerph, 2019, doi:10.3390/ijerph16050840_

Round 1
Reviewer 1 Report
Dear Authors,
The presented topic is relevant and the conducted research is set up in the right direction for the field of environmental design. However, significant improvement is needed to make the research question sharper and the whole structure more coherent. The methodology used isn't properly described.
On-site measurements were conducted to investigate the acoustic territory and sound pressure levels which define the area where participants are stimulated (by sound pressure level?) to participate in the dance. In an urban environment such as the chosen sites, sound pressure level can't be the only measure taken into the account. As suggested in the literature reviewed in the introduction, both background and peak levels must be taken into account. Psychoacoustic parameters such as the perceived loudness, sharpness and tonality are more likely to accurately describe this phenomenon. Moreover, it is unlikely that the intensity of the perceived musical content is the only key factor which indicates whether passers by will join the dance or not. The sound system set up must be better described as the measurement could vary depending on the number and the type of speakers and their position in space (height, directionality, are they close to walls etc.). Sounds made by the dancers might also be relevant but this isn't mentioned.
Two separate locations were chosen to investigate the influence of urban morphology and dominant or planned use. It is not clear whether urban morphology and planned uses are key factors, nor why should it be relevant for an ad-hoc activity such as square dance. This feels implied in the introduction, but it isn't clear. I would suggest to carefully describe this without implying that multifunctional public spaces like streets or squares aren't appropriate for such an activity per se but rather investigate which spatial features such as size, openness or visibility could cause the square dance to interfere with some other activities.
It is not clear whether the square dance at the two analysed locations was spontaneous or initiated by the researchers. If the latter is correct, this should be better described, i.e. the recruitment process, age and gender, etc.
Offsite survey provides clearer results. However, the roles of the on-site measurements and the off-site survey within the whole aren't clear. The research question must be made clearer so that on and off site results could be better interpreted. Further, the offsite online survey should be better described as it is not clear if participants share the common understanding about what square dance is. An on-site survey during a square dance could also be a way to ensure all the participants are describing their perception and opinion of the same set of conditions.
The square dance itself must be better explained in the abstract and the introduction as this recent phenomenon and its origins aren't widely known. For instance, a reader could be led to believe it is a name for any kind of dance being performed at a city square by a group of enthusiasts or amateurs.
Investigating a firmer connection with the field of public health might be beneficial as well, i.e. some measured sound pressure levels are considered harmful but participating in a square dance could still have some positive effects on health.
Title could better describe what the research question is. It is not clear whether the focus is on impacts on soundscapes (both positive and negative domains), or it is about factors causing annoyance.
The diagrams representing the results should be more meaningful.
Author Response
Dear Reviewer,
Thanks for your time and comments. Please see our responses to your suggestions and comments below. We have revised our paper accordingly.
The presented topic is relevant and the conducted research is set up in the right direction for the field of environmental design. However, significant improvement is needed to make the research question sharper and the whole structure more coherent. The methodology used isn't properly described.
Thanks for the comments. The paper has been revised to address the comments. Research questions have been revised and made clearer: 1) What influence does square dance have on the acoustic environment in urban streets? 2) How do gender, age and participation influence the enjoyment of square dancing? 3) Are urban streets perceived appropriate for square dancing to occur? The methods section has been expanded with more detailed descriptions.
On-site measurements were conducted to investigate the acoustic territory and sound pressure levels which define the area where participants are stimulated (by sound pressure level?) to participate in the dance. In an urban environment such as the chosen sites, sound pressure level can't be the only measure taken into the account. As suggested in the literature reviewed in the introduction, both background and peak levels must be taken into account. Psychoacoustic parameters such as the perceived loudness, sharpness and tonality are more likely to accurately describe this phenomenon. Moreover, it is unlikely that the intensity of the perceived musical content is the only key factor which indicates whether passers by will join the dance or not. The sound system set up must be better described as the measurement could vary depending on the number and the type of speakers and their position in space (height, directionality, are they close to walls etc.). Sounds made by the dancers might also be relevant but this isn't mentioned.
Thanks for the comments. Methods section has been expanded to give details of measurements and justify methods used. LAeq was used to understand the rapid changes of sound pressure levels during the dance on both sites. It was compared to the mid-day condition without square dancing onsite which can be used as a reference to a background noise level. In terms of peak levels, in the revised table 2 Amax is also included to see the changes of peak levels reached during the measurements. Reflections of sounds made by the dancers has now been reflected in discussion in the revised document.
Two separate locations were chosen to investigate the influence of urban morphology and dominant or planned use. It is not clear whether urban morphology and planned uses are key factors, nor why should it be relevant for an ad-hoc activity such as square dance. This feels implied in the introduction, but it isn't clear. I would suggest to carefully describe this without implying that multifunctional public spaces like streets or squares aren't appropriate for such an activity per se but rather investigate which spatial features such as size, openness or visibility could cause the square dance to interfere with some other activities.
Thanks for the comments. The urban morphology and planned use have impacts on such activities. Argued in the introduction, sidewalks in urban streets for movements are not planned for large group dancing . One of the question is to see whether people perceive the square dancing appropriate in urban streets and what type of places they perceive appropriate for square dancing to occur. The physical settings of space have been further described in the methods section with supplementary visual information from figure 1, figure 2 and photos in table 1.
It is not clear whether the square dance at the two analysed locations was spontaneous or initiated by the researchers. If the latter is correct, this should be better described, i.e. the recruitment process, age and gender, etc.
Thanks for the suggestion. This has been clarified in the revised document. The square dancing at two sites was spontaneous activities, not initiated by the researcher. It is performed daily by the residents.
Offsite survey provides clearer results. However, the roles of the on-site measurements and the off-site survey within the whole aren't clear. The research question must be made clearer so that on and off site results could be better interpreted. Further, the offsite online survey should be better described as it is not clear if participants share the common understanding about what square dance is. An on-site survey during a square dance could also be a way to ensure all the participants are describing their perception and opinion of the same set of conditions.
Thanks for the comments. Purposes of two methods have been clarified in the revised document. The onsite measurements aim to understand the rapid changes of sound pressure levels during the dance in urban streets, comparing the objective loudness with the noise control regulations in China in urban spaces. Off-site survey is used to gain an understanding on people’s perceptions of square dancing and square dancing music in a general sense. Preferences of music content for square dancing might influence people’s perceptions of the soundscape overall in an in-situ experience. This might be very different from ones’ general opinions on square dancing since the music changes regularly. Participants in the survey are recruited from local social media groups. 78% identified themselves as local residents (living in the case city) who share the same contextual understanding of square dancing asked in the survey. Since square dance is a national-wide practiced daily exercise, the rest 22%(family members of local residents, but living in other cities) should also share similar understanding of what square dancing is as the local residents.
The square dance itself must be better explained in the abstract and the introduction as this recent phenomenon and its origins aren't widely known. For instance, a reader could be led to believe it is a name for any kind of dance being performed at a city square by a group of enthusiasts or amateurs. Investigating a firmer connection with the field of public health might be beneficial as well, i.e. some measured sound pressure levels are considered harmful but participating in a square dance could still have some positive effects on health.
Thanks for the suggestion. In the revised introduction, square dancing phenomenon has been explained regarding to its context, origin and positive impacts on health.
Title could better describe what the research question is. It is not clear whether the focus is on impacts on soundscapes (both positive and negative domains), or it is about factors causing annoyance.
Thanks for the suggestion. Title now has changed to An investigation of the soundscape and enjoyment of square dancing in urban streets: A case study of square dancing in a county level city in China.
The diagrams representing the results should be more meaningful.
Thanks for the suggestion. Figure 3 has been replaced with a more comprehensive chart showing the enjoyment of watching square dancing and annoyance of the music in relation to participants’ age ranges.
Reviewer 2 Report
This paper investigates the influence of the square dance in urban streets on the soundscape through a case study in a county level city in China.
The paper deals only with two sites. It can not be considered exhaustive.
The work as structured can only be considered a technical note
The acoustic measurements were reported in dB, but I think that this value must be reported in dBA. This is a mistake or the acoustic measurements were carried out in dB (Linear)?
The square dance has gained popularity in Chinese cities for physical exercise, what kind of music plays? (rock, pop, ethnic,…)
page 1 row 43 - 70dB - 55dB --- > dB or dBA ?
page 2 row 102:. A-weighted equivalent sound levels (LAeq) will be measured
why only LAeq were reported? The statistical levels as L95 can help to understand the sound field in the square. Why this levels are not reported?
Furthermore the temporal history and the frequency analysis of the acoustic measurements are not reported. These graphs can help you better understand the sound field.
Rows 120-121: The survey was distributed online through local social networks. In total, 106 responses.
Are the authors sure that everyone who answered knows the problem?
Why an acoustic map of the LeqA and L95 has not been reported in order to understand the spatial average distribution in the pizza and to evaluate with the map the area in which the greatest sound level is concentrated?
Row 175 pag 7. Using a Chi-squared test.
Why did the authors use this test? There are other tests. The authors should say the reason for this choice.
If you change tests, the final results change?
Row 233 pag. 9: The authors could insert maps with indications of the singular points to better understand the problem that is being discussed.
Author Response
Dear Reviewer,
Thanks for your time and comments. We have responded to your suggestions and comments accordingly in the revised document. Summary of responses are as below.
This paper investigates the influence of the square dance in urban streets on the soundscape through a case study in a county level city in China.
The paper deals only with two sites. It can not be considered exhaustive.
The work as structured can only be considered a technical note
The acoustic measurements were reported in dB, but I think that this value must be reported in dBA. This is a mistake or the acoustic measurements were carried out in dB (Linear)?
Thanks for pointing out. It is a mistake. This has been thoroughly corrected in the revised document.
The square dance has gained popularity in Chinese cities for physical exercise, what kind of music plays? (rock, pop, ethnic,…)
This has now been clarified in the introduction of square dancing phenomenon: The music varies from folk to pop and the dancing forms vary from gymnastic exercises to folk dance and disco. Often, the moves are easy to follow with the music.
page 1 row 43 - 70dB - 55dB --- > dB or dBA ?
It should be dBA.
page 2 row 102:. A-weighted equivalent sound levels (LAeq) will be measured
why only LAeq were reported? The statistical levels as L95 can help to understand the sound field in the square. Why this levels are not reported?
The aim of the measurement is to understand the rapid changes on sound pressure levels in the streets during the dance rather than exploring the long-term background noise levels. A-weighted equivalent sound levels (LAeq) are be measured at slow-mode for 1-minute intervals three times at three different positions. Thus, it won’t been meaningful to report L95 in this case. Also, for similar purposes, only LAeq was considered in the study conducted by Meng and Kang(2017) about impacts of music related activities on soundscapes in urban spaces.
Furthermore the temporal history and the frequency analysis of the acoustic measurements are not reported. These graphs can help you better understand the sound field.
Thanks for the suggestion. Frequency analysis might be useful if the purpose is to identify types of sounds and spectrums which causes annoyance. However, the authors think this is not the purpose of the measurements.
Rows 120-121: The survey was distributed online through local social networks. In total, 106 responses. Are the authors sure that everyone who answered knows the problem?
Participants in the survey are recruited from local social media groups. 78% identified themselves as local residents (living in the case city) who share the same contextual understanding of square dancing asked in the survey. Since square dance is a national-wide practiced daily exercise, the rest 22%(family members of local residents, but living in other cities) should also share similar understanding of what square dancing is as the local residents
Why an acoustic map of the LeqA and L95 has not been reported in order to understand the spatial average distribution in the pizza and to evaluate with the map the area in which the greatest sound level is concentrated?
Thanks for the suggestion. The author think this will be useful to consider for future investigations on the site specific acoustic investigations using simulation. In this work, it is not included because the purpose was to understand the actual changes sound pressure levels onsite from the speaker to the edge of the dancing group.
Row 175 pag 7. Using a Chi-squared test.
Why did the authors use this test? There are other tests. The authors should say the reason for this choice.If you change tests, the final results change?
The structure of our survey questionnaire was based on 4 key multiple choice questions whose answers were defined as specific categories rather than continuous data. We felt the chi squared test was a useful tool to test simple hypotheses based on age and gender and to examine whether there was an observable significant difference between the expected frequencies and the observed frequencies in these categories. In addition our sample size of 106 was large enough to use the chi-squared as opposed to other tests. The test was predominantly used to study the distribution of categorised data from 4 questions into gender and age bands. Our survey questions were as follows:
Do you enjoy watching Square Dancing – (3 categories of response)
How often do you participate in square dance – (6 categories of response)
Do you find square dance music unpleasant – (4 categories of response)
What places are more appropriate for square dancing – (7 categories of response)
Using the chi squared test we were able to ascertain whether gender or age had a significant effect on the experience of square dance, the enjoyment of the music or the appropriateness of the location by establishing a range of null hypotheses and comparing the categories of data with the expected distribution of data. The chi-squared test allowed us to propose null hypotheses that gender or age have no significant impact on the distribution of preference categories. In the case of gender we were not able to reject the null hypothesis, where as in the case of age were found that the null hypothesis could in fact be rejected. This allowed us to clarify in a simple and clear way that gender has no impact on the preference categories, yet age does have a significant impact.
Row 233 pag. 9: The authors could insert maps with indications of the singular points to better understand the problem that is being discussed.
Thanks for the suggestions. Figure 2 in the document shows the three measuring points in relation to the dancing group and speaker onsite. This has been cross referenced and explained in the revised document.
Round 2
Reviewer 1 Report
Dear Authors,
The manuscript is significantly improved and I believe the new title is much more appropriate. Research questions are made very clear and I believe they are a good choice. Investigating the phenomena using dual perceptual evaluations (participants and others) and objective measurements is an excellent approach. However, I still have some concerns which I believe should be addressed and solved in a more sound manner.
Abstract:
The abstract should be made more efficient. Some minor corrections and rephrasing are needed (specified later).
Throughout the whole paper I would recommend referring to the ISO 12913-1:2014 definition of soundscape and the distinction between the acoustic environment and the soundscape construct, accordingly.
The concluding sentences in the abstract should be rewritten following other amendments.
I believe the method should be more precisely described in the abstract, not only for the offsite survey – specify number of test sites, specify which results relate to the onsite survey, which to the offsite survey, are there any results produced from the both surveys combined? Clearly connecting research questions, methods and results in the abstract would make it very easy to understand and to follow.
Introduction and references:
More references should be used in this section to firmly connect this research to existing theoretical frameworks and recent research. It is especially important to refer to the body of literature considering soundscape and public heath. Daily newspapers shouldn't be the key literature on this. You might look at the paper by Aletta, Oberman and Kang (2018) Associations between positive health-related effects and soundscapes perceptual constructs: A systematic review - for the relation between the two fields and further lead.
For the literature on the use of public space and soundscape, you might consider: Bild, Coler, Pfeffer and Bertolini (2016) Considering Sound in Planning and Designing Public Spaces: A Review of Theory and Applications and a Proposed Framework for Integrating Research and Practice; Bild, Pfeffer, Coler, Rubin and Bertoloni (2018) Public Space Users’ Soundscape Evaluations in Relation to Their Activities. An Amsterdam-Based Study; Estevez-Mauriz, Forssen and Dohmen (2018) Is the sound environment relevant for how people use common spaces?; Astolfi, Orecchia, Bo, Shtrepi, Calleri and Aletta (2018) Influence of Soundscapes on Perception of Safety and Social Presence in an Open Public Space; Aletta, Lepore, Lostara-Konstantinou, Kang and Astolfi (2016) An Experimental Study on the Influence of Soundscapes on People's Behaviour in an Open Public Space.
For the literature considering musical sound sources in public spaces you might consider: Jambrosic, Horvat and Domitrovic (2013) Assessment of urban soundscapes with the focus on an architectural installation with musical features.
For the literature considering psychoacoustics, please see Zwicker and Fastl (1990) Psychoacoustics: Facts and Models.
You might also consider a different theoretical construct than the acoustic territory, as used by LaBelle, and ground the investigation firmer in research on sound propagation, if you find this necessary to elaborate the topic.
Methods:
Although the descriptions are improved, I still have several concerns regarding the methods used - especially regarding the onsite research.
The investigated difference between the mid-day and evening levels isn’t sound enough for the purpose of this study. I would recommend measuring acoustic parameters immediately before and after the square dance. Or perhaps the same location and time of day but on a day when there’s no square dancing going on. Please have in mind, day (Lday) and evening (Levening) levels are considered and approached as different categories in the EU legislative.
Moreover, it isn't described how the judgments on dominance of specific sound sources were made/measured.
Further, both study sites should be better described. They are both considered to be urban streets, yet this is not so clear in the case of Qingjang riverside walk. While it might be argued that there isn’t enough space for square dancing alongside the Binjiang Road, this doesn’t seem to be the case at the Riverside. Further, the size of the dancing groups seem to support this - and it is perhaps more relevant than the volume of the music played.
Some minor issues:
- the term 'folk music' isn't clear enough. If not specified, it might be interpreted as American folk music. Some common psychoacoustic attributes or musical descriptions of the music present in the analysed cases would also be beneficial for understanding the phenomena, i.e. tempo, signature, 'frequency curve', perceived loudness, etc. They could provide better insight in the phenomena than the musical styles mentioned.
- line 10-11. The topic would be even clearer if you would stress that square dancing is mainly an urban outdoor activity.
- line 11 - The expression ‘mid-aged women and elderly people’ is widely used in the paper, yet it might imply that participants are female if they are mid-aged and both male and female if they are elderly. This should be more precise.
- line 47-48 - sentence should be rephrased or a reference added
- line 53 - it is not clear what is meant by 'aesthetical judgments on soundscapes', is it only about engagement and enjoyment? Consider the difference between the acoustic environment and soundscape as per ISO 12913-1:2014 definition
- line 72 - the typological issue should be made clearer
- line 91-92 - reference needed. What about informal use of public space? Cafes can be planned and designed in the streets as well. Further, pedestrian streets and high streets should also be considered when discussing the typology of public spaces.
- line 102-103 - reference needed or please rephrase. 'Certain public spaces' isn't clear enough.
- line 109.110 - reference needed or please rephrase.
- line 122 - 'around120 square', there should be space
- line 122-124 - sentence should be rephrased, otherwise a reference is needed
- line 125 - 'with an paved'
- line 130 - wrong tense
- line 134 - wrong tense
- line 148 - 'instrumental' is not clear, please rephrase
- line 149 - wrong tense
- line 155 - please indicate the date when the survey took place
- line 174 - perhaps consider a newer reference
- line 292 - a stronger reference would be better, if possible
- line 309 - it is not clear what is meant by 'without gender bias', please rephrase
Limitations of the study:
As the topic is novel and important, I would encourage you to make amends to the paper to make it scientifically sound and 'bullet-proof'. I believe that clearly stating the limitations of the study in a dedicated section, instead of rushing to too many conclusions, would greatly benefit the paper.
Here you could discuss perhaps whether speaker directivity or using a blue tooth - based speaker array would influence the spatial organisation of the dancers and the levels.
What physical features of streets were considered to be a limiting factor for square dancing? How is this investigated?
Conclusions:
There is not enough evidence to conclude that the level of the square dance music is the key factor to define the area that will be occupied by the dancers. Square dance is a social activity and it is expected that the decision to participate is more complex. Further, sound waves travel following a different geometry. After all, I don't find this specific conclusion and the 73 dB threshold explanation to be closely related to the research questions.
There is not enough evidence to draw conclusions about the influence about the visual setting ('appearance of the physical space'), i.e. effect of natural lighting, large body of water, foliage etc. I would suggest discussing this in the section describing the limitations of the study.
Title:
You might consider if ‘soundscape and enjoyment’ are best terms for the heading. Perhaps you could you something like 'soundscape factors'? Further, it’s not necessary to mention ‘square dancing’ twice in the heading.
Once again, I believe the topic is of interest for both the community of soundscape researchers and the wider community and it is worthwhile to improve it and publish it.
Author Response
Abstract:
The abstract should be made more efficient. Some minor corrections and rephrasing are needed (specified later).
Throughout the whole paper I would recommend referring to the ISO 12913-1:2014 definition of soundscape and the distinction between the acoustic environment and the soundscape construct, accordingly.
The concluding sentences in the abstract should be rewritten following other amendments.
I believe the method should be more precisely described in the abstract, not only for the offsite survey – specify number of test sites, specify which results relate to the onsite survey, which to the offsite survey, are there any results produced from the both surveys combined? Clearly connecting research questions, methods and results in the abstract would make it very easy to understand and to follow.
Thanks for the suggestion. Abstract has been revised to include more detailed descriptions of methods.
Introduction and references:
More references should be used in this section to firmly connect this research to existing theoretical frameworks and recent research. It is especially important to refer to the body of literature considering soundscape and public heath. Daily newspapers shouldn't be the key literature on this. You might look at the paper by Aletta, Oberman and Kang (2018) Associations between positive health-related effects and soundscapes perceptual constructs: A systematic review - for the relation between the two fields and further lead.
For the literature on the use of public space and soundscape, you might consider: Bild, Coler, Pfeffer and Bertolini (2016) Considering Sound in Planning and Designing Public Spaces: A Review of Theory and Applications and a Proposed Framework for Integrating Research and Practice; Bild, Pfeffer, Coler, Rubin and Bertoloni (2018) Public Space Users’ Soundscape Evaluations in Relation to Their Activities. An Amsterdam-Based Study; Estevez-Mauriz, Forssen and Dohmen (2018) Is the sound environment relevant for how people use common spaces?; Astolfi, Orecchia, Bo, Shtrepi, Calleri and Aletta (2018) Influence of Soundscapes on Perception of Safety and Social Presence in an Open Public Space; Aletta, Lepore, Lostara-Konstantinou, Kang and Astolfi (2016) An Experimental Study on the Influence of Soundscapes on People's Behaviour in an Open Public Space.
For the literature considering musical sound sources in public spaces you might consider: Jambrosic, Horvat and Domitrovic (2013) Assessment of urban soundscapes with the focus on an architectural installation with musical features.
For the literature considering psychoacoustics, please see Zwicker and Fastl (1990) Psychoacoustics: Facts and Models.
You might also consider a different theoretical construct than the acoustic territory, as used by LaBelle, and ground the investigation firmer in research on sound propagation, if you find this necessary to elaborate the topic.
Thanks for the suggestions. Following references has been added to support the argument:
Aletta, F., Oberman, T., & Kang, J. (2018). Associations between positive health-related effects and soundscapes perceptual constructs: A systematic review. International journal of environmental research and public health, 15(11), 2392.
Aletta, F., Lepore, F., Kostara-Konstantinou, E., Kang, J., & Astolfi, A. (2016). An experimental study on the influence of soundscapes on people’s behaviour in an open public space. Applied Sciences, 6(10), 276.
Bild, Pfeffer, Coler, Rubin and Bertoloni (2018) Public Space Users’ Soundscape Evaluations in Relation to Their Activities. An Amsterdam-Based Study.
Bild E., Steele D., Pfeffer K., Bertolini L., Guastavino C. (2018). “Activity as a mediator between users and their auditory environment in an urban pocket park: a case study of Parc du Portugal (Montreal, Canada),” in Handbook of Research on Perception-Driven Approaches to Urban Assessment and Design eds Aletta F., Xiao, J., editors. Hershey, PA: IGI Global. pp.100–125.
ISO 12913-1:2014 Acoustics—Soundscape—Part 1: Definition and conceptual framework. Geneva, Switzerland: International Organization for Standardization.
Estevez-Mauriz, Forssen and Dohmen (2018) Is the sound environment relevant for how people use common spaces?
Warburton, D. E., Nicol, C. W., & Bredin, S. S. (2006). Health benefits of physical activity: the evidence. Cmaj, 174(6), 801-809.
Prato, P. (1984). Music in the streets: the example of Washington Square Park in New York City. Popular Music, 4, 151-163.
National Standards of People's Republic of China (2008) GB22337-2008: Emission standard for community noise.
Methods:
Although the descriptions are improved, I still have several concerns regarding the methods used - especially regarding the onsite research.
The investigated difference between the mid-day and evening levels isn’t sound enough for the purpose of this study. I would recommend measuring acoustic parameters immediately before and after the square dance. Or perhaps the same location and time of day but on a day when there’s no square dancing going on. Please have in mind, day (Lday) and evening (Levening) levels are considered and approached as different categories in the EU legislative.
According to Chinese regulations on emissions of community GB22337-2008, the definition for daytime is 6am-10pm. Square dance is between 6pm and 8pm. There isn’t any difference from the daytime noise control standards. Although agreed it might be useful to also have measurements straight after the dance finishes, the current comparison with mid-day measurements also show the difference square dancing has made to the sound pressure levels on both sites.
Moreover, it isn't described how the judgments on dominance of specific sound sources were made/measured.
As described in methods, the dominant sound were identified through ethnographical observations conducted onsite by the researcher.
Further, both study sites should be better described. They are both considered to be urban streets, yet this is not so clear in the case of Qingjang riverside walk. While it might be argued that there isn’t enough space for square dancing alongside the Binjiang Road, this doesn’t seem to be the case at the Riverside. Further, the size of the dancing groups seem to support this - and it is perhaps more relevant than the volume of the music played.
By definition, both streets are considered as urban streets in the Chinese context. Qingjiang road is located in the new town area of the city which has a different scale in design to fit its political purposes with most government buildings aside. The volume of music played can be indicated through the measuring point a, 1 metre from the speaker. As described in the observation notes table 1, the size of the group grows in the first half an hour. The speaker volume is set at the start of dance and fixed through the whole dance. It is more likely that the edges of the dancing group are determined by the visual distance to the dance leaders and aural perceptions of the music rather than the other way around.
Some minor issues:
- the term 'folk music' isn't clear enough. If not specified, it might be interpreted as American folk music. Some common psychoacoustic attributes or musical descriptions of the music present in the analysed cases would also be beneficial for understanding the phenomena, i.e. tempo, signature, 'frequency curve', perceived loudness, etc. They could provide better insight in the phenomena than the musical styles mentioned.
- line 10-11. The topic would be even clearer if you would stress that square dancing is mainly an urban outdoor activity.
- line 11 - The expression ‘mid-aged women and elderly people’ is widely used in the paper, yet it might imply that participants are female if they are mid-aged and both male and female if they are elderly. This should be more precise.
- line 47-48 - sentence should be rephrased or a reference added
- line 53 - it is not clear what is meant by 'aesthetical judgments on soundscapes', is it only about engagement and enjoyment? Consider the difference between the acoustic environment and soundscape as per ISO 12913-1:2014 definition
- line 72 - the typological issue should be made clearer
- line 91-92 - reference needed. What about informal use of public space? Cafes can be planned and designed in the streets as well. Further, pedestrian streets and high streets should also be considered when discussing the typology of public spaces.
- line 102-103 - reference needed or please rephrase. 'Certain public spaces' isn't clear enough.
- line 109.110 - reference needed or please rephrase.
- line 122 - 'around120 square', there should be space
- line 122-124 - sentence should be rephrased, otherwise a reference is needed
- line 125 - 'with an paved'
- line 130 - wrong tense
- line 134 - wrong tense
- line 148 - 'instrumental' is not clear, please rephrase
- line 149 - wrong tense
- line 155 - please indicate the date when the survey took place
- line 174 - perhaps consider a newer reference
- line 292 - a stronger reference would be better, if possible
- line 309 - it is not clear what is meant by 'without gender bias', please rephrase
Thanks for the suggestions. The above points have been corrected and clarified in the revised documents. Please see the document with tack changes.
Limitations of the study:
As the topic is novel and important, I would encourage you to make amends to the paper to make it scientifically sound and 'bullet-proof'. I believe that clearly stating the limitations of the study in a dedicated section, instead of rushing to too many conclusions, would greatly benefit the paper.
Here you could discuss perhaps whether speaker directivity or using a blue tooth - based speaker array would influence the spatial organisation of the dancers and the levels.
What physical features of streets were considered to be a limiting factor for square dancing? How is this investigated?
Thanks for the suggestion. A section has been added as ‘limitation of the study’ to address the above points.
Conclusions:
There is not enough evidence to conclude that the level of the square dance music is the key factor to define the area that will be occupied by the dancers. Square dance is a social activity and it is expected that the decision to participate is more complex. Further, sound waves travel following a different geometry. After all, I don't find this specific conclusion and the 73 dB threshold explanation to be closely related to the research questions.
There is not enough evidence to draw conclusions about the influence about the visual setting ('appearance of the physical space'), i.e. effect of natural lighting, large body of water, foliage etc. I would suggest discussing this in the section describing the limitations of the study.
Thanks for the suggestion. This has been discussed in the limitation section in the revised document.
Title:
You might consider if ‘soundscape and enjoyment’ are best terms for the heading. Perhaps you could you something like 'soundscape factors'? Further, it’s not necessary to mention ‘square dancing’ twice in the heading.
Thanks for the suggestion. Agreed that ‘soundscape factors’ will be more appropriate for the title. Title has been changed to ‘’
Many thanks,
Jieling and Andy
Reviewer 2 Report
pag. 4 row 149
the peak levels (Amax) .
May be: LAmax
Author Response
Dear Reviewer,
Thanks again for reviewing the paper and providing useful and constructive suggestions. We have made revisions according to your suggestions. Here are responses to your concerns. Please also refer to the revised manuscript and the document with track changes.
Abstract:
The abstract should be made more efficient. Some minor corrections and rephrasing are needed (specified later).
Throughout the whole paper I would recommend referring to the ISO 12913-1:2014 definition of soundscape and the distinction between the acoustic environment and the soundscape construct, accordingly.
The concluding sentences in the abstract should be rewritten following other amendments.
I believe the method should be more precisely described in the abstract, not only for the offsite survey – specify number of test sites, specify which results relate to the onsite survey, which to the offsite survey, are there any results produced from the both surveys combined? Clearly connecting research questions, methods and results in the abstract would make it very easy to understand and to follow.
Thanks for the suggestion. Abstract has been revised to include more detailed descriptions of methods.
Introduction and references:
More references should be used in this section to firmly connect this research to existing theoretical frameworks and recent research. It is especially important to refer to the body of literature considering soundscape and public heath. Daily newspapers shouldn't be the key literature on this. You might look at the paper by Aletta, Oberman and Kang (2018) Associations between positive health-related effects and soundscapes perceptual constructs: A systematic review - for the relation between the two fields and further lead.
For the literature on the use of public space and soundscape, you might consider: Bild, Coler, Pfeffer and Bertolini (2016) Considering Sound in Planning and Designing Public Spaces: A Review of Theory and Applications and a Proposed Framework for Integrating Research and Practice; Bild, Pfeffer, Coler, Rubin and Bertoloni (2018) Public Space Users’ Soundscape Evaluations in Relation to Their Activities. An Amsterdam-Based Study; Estevez-Mauriz, Forssen and Dohmen (2018) Is the sound environment relevant for how people use common spaces?; Astolfi, Orecchia, Bo, Shtrepi, Calleri and Aletta (2018) Influence of Soundscapes on Perception of Safety and Social Presence in an Open Public Space; Aletta, Lepore, Lostara-Konstantinou, Kang and Astolfi (2016) An Experimental Study on the Influence of Soundscapes on People's Behaviour in an Open Public Space.
For the literature considering musical sound sources in public spaces you might consider: Jambrosic, Horvat and Domitrovic (2013) Assessment of urban soundscapes with the focus on an architectural installation with musical features.
For the literature considering psychoacoustics, please see Zwicker and Fastl (1990) Psychoacoustics: Facts and Models.
You might also consider a different theoretical construct than the acoustic territory, as used by LaBelle, and ground the investigation firmer in research on sound propagation, if you find this necessary to elaborate the topic.
Thanks for the suggestions. Following references has been added to support the argument:
Aletta, F., Oberman, T., & Kang, J. (2018). Associations between positive health-related effects and soundscapes perceptual constructs: A systematic review. International journal of environmental research and public health, 15(11), 2392.
Aletta, F., Lepore, F., Kostara-Konstantinou, E., Kang, J., & Astolfi, A. (2016). An experimental study on the influence of soundscapes on people’s behaviour in an open public space. Applied Sciences, 6(10), 276.
Bild, Pfeffer, Coler, Rubin and Bertoloni (2018) Public Space Users’ Soundscape Evaluations in Relation to Their Activities. An Amsterdam-Based Study.
Bild E., Steele D., Pfeffer K., Bertolini L., Guastavino C. (2018). “Activity as a mediator between users and their auditory environment in an urban pocket park: a case study of Parc du Portugal (Montreal, Canada),” in Handbook of Research on Perception-Driven Approaches to Urban Assessment and Design eds Aletta F., Xiao, J., editors. Hershey, PA: IGI Global. pp.100–125.
ISO 12913-1:2014 Acoustics—Soundscape—Part 1: Definition and conceptual framework. Geneva, Switzerland: International Organization for Standardization.
Estevez-Mauriz, Forssen and Dohmen (2018) Is the sound environment relevant for how people use common spaces?
Warburton, D. E., Nicol, C. W., & Bredin, S. S. (2006). Health benefits of physical activity: the evidence. Cmaj, 174(6), 801-809.
Prato, P. (1984). Music in the streets: the example of Washington Square Park in New York City. Popular Music, 4, 151-163.
National Standards of People's Republic of China (2008) GB22337-2008: Emission standard for community noise.
Methods:
Although the descriptions are improved, I still have several concerns regarding the methods used - especially regarding the onsite research.
The investigated difference between the mid-day and evening levels isn’t sound enough for the purpose of this study. I would recommend measuring acoustic parameters immediately before and after the square dance. Or perhaps the same location and time of day but on a day when there’s no square dancing going on. Please have in mind, day (Lday) and evening (Levening) levels are considered and approached as different categories in the EU legislative.
According to Chinese regulations on emissions of community GB22337-2008, the definition for daytime is 6am-10pm. Square dance is between 6pm and 8pm. There isn’t any difference from the daytime noise control standards. Although agreed it might be useful to also have measurements straight after the dance finishes, the current comparison with mid-day measurements also show the difference square dancing has made to the sound pressure levels on both sites.
Moreover, it isn't described how the judgments on dominance of specific sound sources were made/measured.
As described in methods, the dominant sound were identified through ethnographical observations conducted onsite by the researcher.
Further, both study sites should be better described. They are both considered to be urban streets, yet this is not so clear in the case of Qingjang riverside walk. While it might be argued that there isn’t enough space for square dancing alongside the Binjiang Road, this doesn’t seem to be the case at the Riverside. Further, the size of the dancing groups seem to support this - and it is perhaps more relevant than the volume of the music played.
By definition, both streets are considered as urban streets in the Chinese context. Qingjiang road is located in the new town area of the city which has a different scale in design to fit its political purposes with most government buildings aside. The volume of music played can be indicated through the measuring point a, 1 metre from the speaker. As described in the observation notes table 1, the size of the group grows in the first half an hour. The speaker volume is set at the start of dance and fixed through the whole dance. It is more likely that the edges of the dancing group are determined by the visual distance to the dance leaders and aural perceptions of the music rather than the other way around.
Some minor issues:
- the term 'folk music' isn't clear enough. If not specified, it might be interpreted as American folk music. Some common psychoacoustic attributes or musical descriptions of the music present in the analysed cases would also be beneficial for understanding the phenomena, i.e. tempo, signature, 'frequency curve', perceived loudness, etc. They could provide better insight in the phenomena than the musical styles mentioned.
- line 10-11. The topic would be even clearer if you would stress that square dancing is mainly an urban outdoor activity.
- line 11 - The expression ‘mid-aged women and elderly people’ is widely used in the paper, yet it might imply that participants are female if they are mid-aged and both male and female if they are elderly. This should be more precise.
- line 47-48 - sentence should be rephrased or a reference added
- line 53 - it is not clear what is meant by 'aesthetical judgments on soundscapes', is it only about engagement and enjoyment? Consider the difference between the acoustic environment and soundscape as per ISO 12913-1:2014 definition
- line 72 - the typological issue should be made clearer
- line 91-92 - reference needed. What about informal use of public space? Cafes can be planned and designed in the streets as well. Further, pedestrian streets and high streets should also be considered when discussing the typology of public spaces.
- line 102-103 - reference needed or please rephrase. 'Certain public spaces' isn't clear enough.
- line 109.110 - reference needed or please rephrase.
- line 122 - 'around120 square', there should be space
- line 122-124 - sentence should be rephrased, otherwise a reference is needed
- line 125 - 'with an paved'
- line 130 - wrong tense
- line 134 - wrong tense
- line 148 - 'instrumental' is not clear, please rephrase
- line 149 - wrong tense
- line 155 - please indicate the date when the survey took place
- line 174 - perhaps consider a newer reference
- line 292 - a stronger reference would be better, if possible
- line 309 - it is not clear what is meant by 'without gender bias', please rephrase
Thanks for the suggestions. The above points have been corrected and clarified in the revised documents. Please see the document with tack changes.
Limitations of the study:
As the topic is novel and important, I would encourage you to make amends to the paper to make it scientifically sound and 'bullet-proof'. I believe that clearly stating the limitations of the study in a dedicated section, instead of rushing to too many conclusions, would greatly benefit the paper.
Here you could discuss perhaps whether speaker directivity or using a blue tooth - based speaker array would influence the spatial organisation of the dancers and the levels.
What physical features of streets were considered to be a limiting factor for square dancing? How is this investigated?
Thanks for the suggestion. A section has been added as ‘limitation of the study’ to address the above points.
Conclusions:
There is not enough evidence to conclude that the level of the square dance music is the key factor to define the area that will be occupied by the dancers. Square dance is a social activity and it is expected that the decision to participate is more complex. Further, sound waves travel following a different geometry. After all, I don't find this specific conclusion and the 73 dB threshold explanation to be closely related to the research questions.
There is not enough evidence to draw conclusions about the influence about the visual setting ('appearance of the physical space'), i.e. effect of natural lighting, large body of water, foliage etc. I would suggest discussing this in the section describing the limitations of the study.
Thanks for the suggestion. This has been discussed in the limitation section in the revised document.
Title:
You might consider if ‘soundscape and enjoyment’ are best terms for the heading. Perhaps you could you something like 'soundscape factors'? Further, it’s not necessary to mention ‘square dancing’ twice in the heading.
Thanks for the suggestion. Agreed that ‘soundscape factors’ will be more appropriate for the title. Title has been changed to ‘’
Many thanks,
Jieling and Andy
Round 3
Reviewer 1 Report
Dear Authors,
Thank you for taking your time to address my comments. Several points are now made much clearer.
However, I believe there has been no significant improvement in the quality after this process.
Therefore, I believe the best way to proceed is to ask for another reviewer who will offer a fresh insight into the topic.
Kind regards.
Author Response
Dear Editor,
Thank you for the time and valuable advice. We have revised the manuscript to address your comments.
The English is generally fine but a further proof-reading would be beneficial as I have spotted several typos and or inconsistencies. For example: off-site/off site not consistent throughout the text; T-test chi-square/chi square; last sentences of the abstract, grammar to revise (“the results found no gender…”); square dance/square dancing; music-associated/music associated; everyday instead of every day; “SLM WAS provided” (instead of will be); check the tenses in general; Amax instead of LAmax in caption of Table 2; etc. Please revise the text carefully, as there are several others I noticed and did not mention. Also, the formatting of the Tables should be revised as it is not ideal at the moment.
Typos have been checked and corrected.
In the Introduction, I would ask the authors to slightly revise the narrative, as the transition from one paragraph/argument to another is not always smooth. It is not clear from this section whether the authors want to stress the importance of the work on the measurements part or rather the sociological (and possibly public health) aspects. At the end of the Intro, the reader should be able to see where the paper is going. It is OK to keep content concise, because this is submitted as brief report, but please revise slightly, having in mind that the readership of this journal is interested in “environmental research and public health”.
The introduction has been restructured to justify the relation of this paper to public health.
Section 2.2 – the 1-minute interval is not very common in environmental acoustics monitoring: please add some rationale to justify such a short measurement period for this specific square dance application. Also, provide justification for using LAeq and LAmax in particular (use references if needed).
This has been refelected. A reference has been added:
Kang, J., Aletta, F., Margaritis, E., & Yang, M. (2018). A model for implementing soundscape maps in smart cities. Noise Mapping, 5(1), 46-59.
Section 2.3 should be improved a bit – the age groups could be plotted as a figure, with a histogram for the distribution. If available, mean age and standard deviation should be reported. There is some repetition: “an age range from twenty years old to seventy” and then same info is repeated a few lines below with numbers.
This has been revised. The participants for the survey only asked to choose their age range but not give exact age.
The questionnaire/protocol is not reported in big detail and at this stage this is somewhat confusing. For instance: where the 5 questions designed for open answers? Were there options for reply?
This has been clarified. Participants were given options to choose from for each question. Also, a number of tables have been added to show examples of T-tests and Chi-squared tests conducted which also shows type of options given to participants.
For the psycho-acoustic descriptors: I would rather call them soundscape descriptors as psychoacoustics refers mostly to other parameters (sharpness, loudness, etc.). Also it is not clear on what scales these were assessed. How were the annoying-pleasing, noisy-rhythmic, boring-interesting? Please add some references.
References were added:
Axelsson, Ö., Nilsson, M. E., & Berglund, B. (2010). A principal components model of soundscape perception. The Journal of the Acoustical Society of America, 128(5), 2836-2846.
Maculewicz, J., Erkut, C., & Serafin, S. (2016). How can soundscapes affect the preferred walking pace?. Applied Acoustics, 114, 230-239.
Davies, W. J., Adams, M. D., Bruce, N. S., Cain, R., Carlyle, A., Cusack, P., ... & Marselle, M. (2013). Perception of soundscapes: An interdisciplinary approach. Applied acoustics, 74(2), 224-231.
Section 3.1 - Sometimes the authors refer to “loudness” throughout the text, but there is a risk for this word to be misinterpreted as the psychoacoustic loudness of Fastl and Zwicker. When possible, please replace with sound level, or “magnitude of the auditory stimulus”, or similar.
In the Table 2, please use only 1 digit for the decimals of the dB levels. Please better format the table to improve readability; please revise caption: LAmax, not Amax.
Corrected.
Section 3.2 - The statistics reported in here is a bit hard to read and interpret. It is not clear which tests were used on which variables (and whether the tests were appropriate, met the statistical assumptions, etc.). The American Psychological Association has clear guidelines on how to report statistics. I recommend the authors have a look and change the session to report in a standardized way (i.e., APA style).
Corrected.
Discussion – the title of subsection 4.1 is not very clear. A reference is not reported as number here, Labelle (2010).
Reference has been labelled. 4.1 Section title has changed to Acoustic boundaries between square dancers and audience in streets.
I would move the limitations of the study (5.1) to Discussion, suppress the heading of 5.2 and leave key findings under conclusions directly.
Restructured as suggested.
Finally, it is important that the authors report the full details (date, number of letter, authority, etc.) of the Ethical Committee from BCU that reviewed and approved this study. This is a requirement of the journal and should be reported in the methods. See the instructions for author of the journal for the exact statement to report.
The Ethical information has been added to methods.
Kind regards,
Jieling and Andy